# Antigiardial Activity of Acetylsalicylic Acid Is Associated with Overexpression of HSP70 and Membrane Transporters

**DOI:** 10.3390/ph13120440

**Published:** 2020-12-03

**Authors:** Verónica Yadira Ochoa-Maganda, Itzia Azucena Rangel-Castañeda, Daniel Osmar Suárez-Rico, Rafael Cortés-Zárate, José Manuel Hernández-Hernández, Armando Pérez-Rangel, Natalia Chiquete-Félix, Gloria León-Ávila, Sirenia González-Pozos, Jorge Gaona-Bernal, Araceli Castillo-Romero

**Affiliations:** 1Departamento de Fisiología, Centro Universitario de Ciencias de la Salud, Universidad de Guadalajara, Sierra Mojada 950, Col. Independencia, Guadalajara, Jalisco 44340, Mexico; veronica.ochmag@gmail.com (V.Y.O.-M.); dosuarezr94@gmail.com (D.O.S.-R.); 2Departamento de Biología Celular, Centro de Investigación y Estudios Avanzados del Instituto Politécnico Nacional, Av. Instituto Politécnico Nacional 2508, Col. San Pedro Zacatenco, Ciudad de México 07360, Mexico; itzia.rangel2591@gmail.com (I.A.R.-C.); manolo@cell.cinvestav.mx (J.M.H.-H.); rangelarm@yahoo.com (A.P.-R.); 3Departamento de Microbiología y Patología, Centro Universitario de Ciencias de la Salud, Universidad de Guadalajara, Sierra Mojada 950, Col. Independencia, Guadalajara, Jalisco 44340, Mexico; rcortesz@hotmail.com (R.C.-Z.); jgaber2007@gmail.com (J.G.-B.); 4Instituto de Fisiología Celular, Universidad Nacional Autónoma de México, Circuito Exterior s/n Ciudad Universitaria, Coyoacán, Ciudad de México 04510, Mexico; nchiquete@ifc.unam.mx; 5Departamento de Zoología, Escuela Nacional de Ciencias Biológicas, IPN, Carpio y Plan de Ayala S/N, Casco de Santo Tomás, Ciudad de México 11340, Mexico; leonavila60@yahoo.com.mx; 6Unidad de Microscopía Electrónica LaNSE, Centro de Investigación y Estudios Avanzados del Instituto Politécnico Nacional, Av. Instituto Politécnico Nacional 2508, Col. San Pedro Zacatenco, Ciudad de México 07360, Mexico; sgonzale@cinvestav.mx

**Keywords:** *Giardia lamblia*, acetylsalicylic acid, heat shock protein 70, membrane transporter

## Abstract

*Giardia lamblia* is a flagellated protozoan responsible for giardiasis, a worldwide diarrheal disease. The adverse effects of the pharmacological treatments and the appearance of drug resistance have increased the rate of therapeutic failures. In the search for alternative therapeutics, drug repositioning has become a popular strategy. Acetylsalicylic acid (ASA) exhibits diverse biological activities through multiple mechanisms. However, the full spectrum of its activities is incompletely understood. In this study we show that ASA displayed direct antigiardial activity and affected the adhesion and growth of trophozoites in a time-dose-dependent manner. Electron microscopy images revealed remarkable morphological alterations in the membrane, ventral disk, and caudal region. Using mass spectrometry and real-time quantitative reverse transcription (qRT-PCR), we identified that ASA induced the overexpression of heat shock protein 70 (HSP70). ASA also showed a significant increase of five ATP-binding cassette (ABC) transporters (giABC, giABCP, giMDRP, giMRPL and giMDRAP1). Additionally, we found low toxicity on Caco-2 cells. Taken together, these results suggest an important role of HSPs and ABC drug transporters in contributing to stress tolerance and protecting cells from ASA-induced stress.

## 1. Introduction

*Giardia lamblia* is a ubiquitous protozoan that colonizes the human upper small intestine, causing an acute and chronic diarrheal disease worldwide, giardiasis. The most commonly used medications for the treatment of this disease are metronidazole (MTZ), tinidazole, nitazoxanide (NTX) and albendazole (ABZ) [1,2,3]. Even though current therapies have proven to be useful, all of them present variable efficacies and adverse side effects. Additionally, drug resistance is an increasing concern [4,5], so much that the search for new safe and effective treatments continues to be a very important issue in experimental and clinical research.

The discovery and development of new drugs are long (10–15 years), complex and expensive processes, and the success rate is only 2.01%. In this regard, drug repurposing is a highly efficient strategy to find new indications for existing approved drugs that have already passed preclinical and clinical stages [6,7], and it stands out as an attractive option to find alternative antigiardial options. In this context, acetylsalicylic acid (ASA) has emerged as an interesting option for repurposing drugs against parasites. ASA may also be used for secondary prevention of stroke and acute cardiac events. ASA acts on the two cyclooxygenase isoforms (COX-1 and COX-2) via acetylation of serine 532 in the active site; inhibits the action of both enzymes and prevents the formation of prostaglandins from arachidonic acid [8]. Epidemiological and clinical studies showed that ASA also reduces the incidence of epithelial tumors by the acetylation of multiple proteins including transcription factors, cytoskeleton proteins, stress response proteins (including the heat shock proteins, HSPs), membrane proteins, among others [9,10,11], suggesting other mechanisms and molecular targets. Some studies have evidenced antiparasitic activity. Aaina and Sushma (2010) have shown that ASA improved the antifilarial activity of diethylcarbamazine [12]. On *Entamoeba histolytica*, it affects the dynamics of the actin cytoskeleton, and amebic movement decreases [13].

On the other hand, several studies have shown that ASA modifies the expression of distinct multidrug-resistant genes, interfering with their transporter activity and generating diversity in the multidrug-resistant phenotype [14,15,16]. ATP-binding cassette (ABC) transporters constitute one of the largest families of integral membrane proteins. In protozoa, they mediate ATP-dependent transport of a wide variety of chemotherapeutic drugs, away from their targets inside the parasites [17]. Another important molecule, HSP70, plays a crucial role in tissue defense mechanisms due to its chaperon activity [18]. In this work, we demonstrated that ASA exposure to *Giardia* affected the growth and adhesion of trophozoites. Microscopy images revealed dramatic changes on membrane and cell morphology. Remarkably, ASA activity was associated with the modulation of HSP70 and the overexpression of five ABC transporters.

## 2. Results

### 2.1. ASA Inhibits the Growth, Adhesion and Cell Viability of Giardia lamblia Trophozoites

The effect of ASA on parasite growth was kinetically determined. Reduction in parasite number was observed 12 h after the assay was initiated (Figure 1A), and IC_50_ value was 0.29 mM at 24 h (Table 1). The maximal inhibitory effects were observed after 48 h of incubation. ASA at 0.25 mM decreased cell growth by 73.9%, whereas treatment with 0.5 and 1 mM decreased cell growth by 90.4% and 98.6%, respectively (Figure 1B), suggesting a time-dose-dependent effect. All assayed ASA concentrations promoted the detachment of trophozoites; parasites showed weak cell surface adhesion depending on dose and time. The adherence was reduced to 20%, 35%, 44% and 76%, with ASA 0.125, 0.25, 0.5 and 1 mM, respectively, after 48 h of incubation (Figure 1C). Finally, viable parasites after exposure to ASA were determined by trypan blue exclusion test. ASA at 0.125, 0.25, 0.5 or 1 mM inhibited cell viability 12 h after the culture started (5%, 5%, 19% and 36%, respectively). The most dramatic effect was observed with 0.5 and 1 mM at 48 h; only 42% and 24% of trophozoites were viable (Figure 1D). Positive control with 1 μM MTZ inhibited trophozoite growth by 24%, 45.7% and 76.5% at 12, 24 and 48 h, respectively. DMSO-treated cells did not exhibit any significant differences compared with untreated cells.

### 2.2. ASA Alters the Morphology of Giardia lamblia Trophozoites

Analyses of ultrastructural changes in trophozoites after 48 h of treatment were performed using scanning electron microscopy. The micrographs revealed untreated and DMSO-treated trophozoites with normal morphology; ventral disc, flagella and ventrolateral flange without alterations (Figure 2A,B), while ASA-treated parasites showed visible morphological alterations, mainly in the membrane and the anterior regions of the trophozoites. With 0.125 and 0.25 mM of ASA, visible alterations in flagella, protrusions and blebs on dorsal and ventral surfaces were observed (Figure 2C,D). ASA at 0.5 and 1 mM substantially reduced the number of parasites and a large amount of cell debris formation was observed; the remnant cells showed membrane he rupture and perforations on the ventral disk (Figure 2E,F).

### 2.3. Tubulin Expression Is Not Modified by ASA Treatment

To address if morphological alterations by ASA might be associated to microtubule destabilization, levels of soluble and polymerized tubulin were detected by Western blotting in parasites exposed to the drug for 24 and 48 h. The results showed similar tubulin expression levels, neither drug concentration nor the time of exposition influenced soluble and insoluble tubulin fractions (Figure 3A,B). When data were expressed as percentage of polymerized tubulin, a slight decrease of protein in the soluble fraction was observed with 0.125 mM of ASA; however, this was not significant (Figure 3C,D).

### 2.4. HSP70 Is Associated with ASA Damage in Giardia

Following electrophoretic separation of *Giardia* extracts, the whole protein pattern after ASA treatment was almost identical to that of the controls, except for one protein ranging from 55 and 70 kDa that was more abundant with increasing concentrations of ASA. The latter was more evident with 0.5 and 1 mM at 48 h (Figure 4A,B). To know the identity of this protein, it was processed by tryptic digestion and UPLC-TOF-MS mass spectrometry was used. Figure 5 shows the obtained mass spectrum of the tryptic peptides. Eleven proteins with more than 96% reliability were identified, among them, 8 corresponded to uncharacterized proteins. The cytosolic HSP70, phosphomannomutase 2, arginine deiminase and Bip had higher sequence coverage of 49.4%, 35.9%, 31.4% and 30.1%, respectively. The other identified proteins with lower coverage were 21.1 protein, ATPase, HSP90, glucose-6-phosphate isomerase and 2 hypothetical proteins (Table 2).

### 2.5. ASA Induced Hsp70 Overexpression in Giardia lamblia Trophozoites

To determine whether ASA regulates *hsp70* expression on *Giardia*, parasites were exposed to 0.5 mM ASA or DMSO, the diluent, for 24 and 48 h, and the expression of *hsp70* was analyzed by SYBR Green Real-Time Quantitative RT-PCR. After ASA treatment, HSP70 mRNA level began to increase after 24 h, reaching a nearly sixfold maximum expression level at 48 h compared with the DMSO control (Figure 6).

### 2.6. ASA Modifies the Expression Level of ABC/MDR Transporter Genes

In mammal cells, it has been demonstrated that some multidrug resistance (MDR) transporters play a critical role in cell death induced by ASA treatment [14,16]. Several works suggest that heat shock factor 1 (HSF1) induces multidrug resistance; it has been related to MDR1 expression and in establishing P-gp transporters, suggesting that HSPs and MDRs could be regulated simultaneously by a different mechanism [19,20,21,22]. To determine whether the response to stress generated in *Giardia* by treatment with ASA is also related to MDR gene overexpression, first, we performed BLAST analysis in the *Giardia* genome database using human P-gp/MDR1, MRP4 and ABCG2 mRNA sequences [14,15,16]. Based on Bit score, sequence identity and e-value, five sequences were identified: ABC transporter family protein (giABC), ABC transporter, putative (giABCP), multidrug resistance ABC transporter ATP-binding and permease protein (giMDRP), MRP-like ABC transporter (giMRPL) and multidrug resistance-associated protein 1 (MDRAP1) (Table 3)

Relative-quantitative RT-PCR analysis demonstrated that all five ABC transporter proteins were differentially overexpressed due to ASA treatment (Figure 7). Interestingly, mRNA expression levels of giABC, giABCP, giMDRP and giMRPL in the first 12 h were significantly higher than in the DMSO control. Subsequent measurements revealed no differences between MDRs and DMSO at 24 h, followed by significantly increased levels at 48 h (Figure 7A–D). At this time, giMDRP reached a maximum expression level nearly twofold than that observed with DMSO, followed by giMRPL (1-fold), giABCP (0.7-fold) and giABC (0.3-fold). For giMDRAP1, a tendency to increase the expression level over time was observed (Figure 7E).

### 2.7. ASA Is Partially Cytotoxic to Caco-2 Cells at High Doses

The cytotoxicity of ASA to human cells was evaluated to support its potential use against *Giardia lamblia*. Caco-2 cells were treated with ASA 0.5 mM for 24 h and, to evaluate viability, an MTT assay was used. Figure 8 shows that ASA at 0.125 and 0.25 mM was not cytotoxic against Caco-2 cells, except for 0.5 and 1 mM of ASA (24.2% and 37.7%, respectively, * *p* < 0.05), with a CC_50_ value of 2.32 mM (Table 1).

## 3. Discussion

Giardiasis is a globally distributed diarrheal disease. There are six classes of compounds whose efficacies are well-studied and that have been approved for the treatment of this infection [23,24]. However, due to undesirable side effects, resistance and an alarming increase of refractory cases to first-line treatments [4,5,25], the search for new alternatives with greater efficacy is still important. Drug repurposing of an approved drug is among the most advantageous strategies to identify new therapeutic alternatives against parasitic protozoan diseases.

Acetylsalicylic acid, commonly named Aspirin, is one of the best documented medicines in the world and is one of the most used drugs of all time. It is the most prescribed anti-inflammatory and analgesic-antipyretic agent [26]. In addition, it has been widely studied to demonstrate new beneficial, biological anticancer, antibacterial or antiparasitic compound properties [13,27,28,29,30]. In this study, the effect of ASA against the parasite *G. lamblia* was demonstrated with an IC_50_ of 0.29 mM, and a SI of 8 (Table 1). Based on SI result, it indicates a high degree of cytotoxic selectivity against *Giardia* with minimal cell toxicity, however, its therapeutic use as an antigiardial candidate is unclear; the SI value is low compared to drugs of therapeutic use [31,32,33]. Even though the effective doses of ASA against *Giardia* are higher than MTZ, it is important to consider that they are different classes of drugs. Additionally, depending on the source, ASA doses range from 75 to 650 mg (approximately 0.4–3.6 mM) for pain relieve or cardiovascular disease. As an antirheumatic drug ASA could be used at higher doses in a range of 3.3 to 24.9 mM/day (http://www.arthritis.org), indicating that it is very often a well-tolerated drug [34,35], thus, ASA shows credible promise as an alternative antigiardial treatment. After ASA treatment, SEM images revealed dramatic alterations on the cell membrane and cytoskeletal structures, which can be associated with the decrease of its adhesion capacity (Figure 1D). Given that microtubules are an essential part of the *Giardia* cytoskeleton [36], soluble and insoluble tubulin was determined by Western blot using a monoclonal anti-α-tubulin antibody, revealing that ASA does not affect tubulin dynamics. According to different investigations, ASA locally perturbs lipid bilayers in a concentration-dependent manner by primarily interacting with lipid head groups and cholesterol, thereby inhibiting raft formation [37,38]. Considering that lipids and fatty acids play an important role in regulating the growth and encystation of *Giardia* and that cholesterol is the only sterol present in trophozoites [39,40], our results suggest that the ASA mechanism of death occurred principally by altering the composition of parasite membrane and causing loss of its integrity as we observed in the SEM micrographs (Figure 2).

On the other hand, prokaryotic and eukaryotic cells respond to potentially harmful stimulations by inducing the synthesis of stress proteins; the heat shock proteins (HSPs) and other metabolites [41]. Using UPLC-TOF-MS mass spectrometry, we identified HSP70, among other stress-related proteins (Table 2). A variety of works have reported that the overexpression of these proteins correlated to resistance to apoptosis induced by a wide range of stimuli [18]. Additional studies are necessary to assess the possible involvement of HSP70 on *Giardia* apoptosis.

Additionally, in *Giardia*, one of the mechanisms related to drug resistance seems to be through ABC membrane transporters [17]. *Giardia* strains resistant to MTZ showed overexpression of different ABC transporters, as well as negative regulation of others, suggesting a possible role of ABC transporters in protecting parasites against this drug [42]. Previous reports have shown that ASA has activity on human Pgp/MDR1, MRP4 and ABCG2 genes [14,15,16]. The search for homologous sequences in *Giardia* DB revealed five ABC transporters with 25% to 35% identity with the query sequences. A time-dependent change was observed in the expression of all the transporters. In contrast to that of giMDRAP1, mRNA expression of giABC, giABCP, giMDRP and giMRPL showed a significant increase at 12 h, but returned to normal levels at 24 h. These changes in ABC transporter expression might be due to morphological changes in the cell membrane caused by ASA. The transporters provide a defense to ASA treatment but eventually fail to afford protection during trophozoites proliferation. This is the first study to provide evidence that ASA kills *Giardia lamblia* trophozoites and, as in mammal cells, the effect of the drug was accompanied by variations in mRNA expression of important molecules such as ABC transporters and HSP70 proteins, necessary for defense against cellular stress. Additional studies are required to establish its mechanism of action.

## 4. Materials and Methods

### 4.1. Giardia Lamblia Trophozoites Culture

Trophozoites of *Giardia lamblia* (WB clone C6) were grown axenically at 37 °C in borosilicate culture tubes containing Diamond’s TYI-S-33 medium, pH 7.0 (supplemented with 0.5 mg/mL of bovine bile and 10% fetal bovine serum) [43]. Cultures were maintained by subculturing the cells twice a week.

### 4.2. Growth Inhibition Assay

To evaluate the effect of ASA on *Giardia lamblia* growth, 10,000 parasites/mL were grown in TYI-S-33 medium containing 0.125, 0.25, 0.5 or 1 mM of ASA (Sigma-Aldrich, Saint Louis, MO, USA), and incubated for 12, 24 and 48 h. Untreated cells and the drug diluent, 0.09% dimethyl sulfoxide (DMSO, Sigma-Aldrich, Saint Louis, MO, USA) were used as negative controls, while 1 µM of metronidazole (MTZ) was used as a positive control. After the incubation periods, the cells were harvested by cooling and counted using a Neubauer chamber. To determine if ASA could affect the pH level, after cell counting, pH was monitored, showing a slight increase from 7.01 to 7.05. The percentage of parasite growth inhibition was calculated in relation to the negative control, which was defined as 100% parasite growth.

### 4.3. Cell Viability Assay

A dye exclusion test was used to determine the viability of trophozoites after DMSO or ASA treatment, about 10 μL of culture was mixed with 10 μL trypan blue 0.4% (Gibco-BRL, Gaithersburg, MD, USA). The total number of parasites (including those which had excluded the dye), was counted in a Neubauer chamber and cell viability was calculated as the percentage of viable cells in the samples relative to untreated cells.

### 4.4. Adherence Assay

To evaluate the effect of ASA on adherence, 10,000 parasites/mL were grown at concentrations and time described above. After the incubation periods, the medium containing nonadherent cells was removed and kept on ice; tubes were filled with cold phosphate-buffered saline (PBS) and placed on ice for 30 min to detach the adherent trophozoites. The number of adherent and nonadherent trophozoites was determined by counting in a Neubauer chamber. The effect on adherence was expressed as percentage of adhered trophozoites in relation to the total number of cells, and the results obtained were compared with control cultures.

### 4.5. Scanning Electron Microscopy (SEM)

To analyze the morphology of trophozoites after ASA or DMSO treatment, parasites were ice-chilled for 20 min, harvested by centrifugation for 10 min at 1973× *g* (4 °C), washed with PBS, fixed for 1 h with 2.5% glutaraldehyde (Sigma-Aldrich, Saint Louis, MO, USA) in PBS and adhered to polyethylenimine-coated coverslips (Sigma-Aldrich, Saint Louis, MO, USA). The fixed cells were washed three times with PBS and post-fixed for 1 h in 1% OsO_4_. Next, cells were washed with PBS, dehydrated in gradient of ethanol series (50–100%) and subjected to critical point drying with CO_2_. Finally, cells were mounted on stainless steel holders, sputter-coated with a thin layer of gold and examined in a JEOL-JSM6510LV SEM (JEOL, Tokyo, Japan).

### 4.6. Preparation of Protein Fractions and Western Blotting

*G. lamblia* trophozoites grown in the presence and absence of ASA under the previously mentioned conditions were used to extract the soluble (cytosolic) and insoluble (polymerized) tubulin fractions according an earlier report [44]. Briefly, the parasites were collected by cooling and posterior centrifugation at 1973× *g* for 10 min at 4 °C. The cell pellets were suspended in lysis buffer (50 mM Tris-Cl, pH 7.4, 150 mM NaCl, 1 mM EDTA, 1% Triton X-100 and Complete Protease Inhibitor (Roche)) and incubated on ice for 30 min. The supernatant (soluble fraction) was obtained by centrifugation at 13,000× *g* at 4 °C. To collect the insoluble fraction, the remaining pellets were resuspended again in lysis buffer and kept at 4 °C. The protein concentration of each fraction was determined by Bradford assay using the Pierce Detergent Compatible Bradford Assay Kit (Thermo Scientific, Waltham, MA, USA) reagent. Readings were made at 595 nm in a microplate reader (Bio-Tek Synergy HT, Winooski, VT, USA). Soluble and polymerized tubulin were then analyzed by sodium dodecyl sulfate-polyacrylamide gel electrophoresis (SDS-PAGE) (10%) and Western blot. Samples (15 μg) were prepared by the addition of 2× Laemmli Sample Buffer (Bio-Rad) supplemented with 5% 2-mercaptoethanol and heated at 100 °C for 5 min in a StableTemp thermoblock (Cole-Parmer, Vernon Hills, IL, USA). The electrophoretic separation of the proteins was carried out at a constant voltage of 100 V for 2 h, gels were either stained with Coomassie blue or transferred to PVDF membranes (Amersham Pharmacia Biotech, Little Chalfont, UK) at a 100 V for 70 min with a Transblot apparatus (Bio-Rad, Hercules, CA, USA). After transfer, the membranes were blocked for 1 h with 5% solution of low-fat milk in PBS supplemented with 0.05% Tween-20 (PBS-T). Three washes with PBS-T were performed and the membranes were incubated for 2 h with 1/200 mouse anti-α-tubulin antibody (Invitrogen, #13-8000, Carlsbad, CA, USA). Subsequently, five washes were performed with 0.05% PBS-T and incubated for 1 h with 1/10,000 goat anti-mouse IgG antibody coupled to horseradish peroxidase (Pierce, #31437, Waltham, MA, USA). The membranes were thoroughly washed with PBS-T and the signal was detected by chemiluminescence (ECL immobilon Western, Millipore, #170-5060, Burlington, MA, USA), the signal was captured with the C-Digit system. Semiquantitative evaluation was performed by densitometry using Image Studio Digits Software version 5.2.

### 4.7. Protein Extract and SDS-PAGE

Trophozoites treated with DMSO or ASA (0.125, 0.25, 0.5 or 1 mM), were collected by centrifugation, suspended in PBS (complemented with protease inhibitors) and lysed by sonication for 30 s at 130 W (3 cycles) (Ultrasonic processor, Sonics and material INC). Cell debris were removed by centrifugation (10,000× *g* for 10 min) and protein concentration was determined by a micro-Bradford assay. Total protein extracts (15 μg) were separated by electrophoresis in 10% (*w*/*v*) SDS-PAGE at 100 V for 2 h. The gel was stained with Coomassie Blue dye to reveal the protein bands. A prominent single major band (55–70 kDa) was excised and processed for mass spectrometry analysis.

### 4.8. Tryptic Digest Protocol Subsequent to Coomassie Staining

For mass spectrometry, the selected band was cut in smaller pieces followed by subsequent destaining with 50% methanol 5% acetic acid solution and rinsed with deionized water. After that, the gel pieces were incubated in 100 mM ammonium bicarbonate (NH_4_HCO_3_). To reduce disulfide bonds and alkylate free cysteines, the gel pieces were incubated for 45 min at 55 °C in 50 mM dithiothreitol (DTT), this solution was then exchanged with 30 mM iodoacetamide (IAA), and the gel pieces were incubated for 30 min at room temperature in the dark. The gel pieces were once more washed with 100 mM NH_4_HCO_3_ and dehydrated with 100% acetonitrile for 10 min. The proteins in the gel pieces were digested with porcine trypsin (20 ng/μL) at 37 °C for 18 h. The resulting peptides were extracted with 50% acetonitrile (*v/v*) and 5% formic acid (*v*/*v*) for 30 min and vacuum dried. Dried peptides were dissolved in 1% formic acid (20 µL), desalted, and concentrated by ZipTip C_18_.

### 4.9. Ultra-Performance Liquid Chromatography-Time of Flight Mass Spectrometry (UPLC-TOF/MS)

All chromatographic measurements were performed on a nanoAcquity UPLC system. The peptides were analyzed in a Waters nanoACQUITY UPLC HSS T3 C_18_ column (1.8 μm ∗ 75 μm ∗ 150 mm). Experimental conditions were as follows: solvent A, 100% water in 0.1% formic acid; solvent B, 100% acetonitrile in 0.1% formic acid, flow rate 0.4 nL/min, column temperature 35 °C, gradient program: 00.00 min 7% B, 54.67 min 40% B, 56.33 min 85% B, 59.64 min 85% B and 61.30 min 7% B. Mass spectrometry analysis was performed on a Waters Synapt G_2_-S TOF spectrometer with an electrospray ionization interface (ESI). Data were collected from positive ions from 50 to 2000 Da with a 1 s scan time. Electrospray ionization conditions were as follows: source temperature, 70 °C; collision energy, 15–45 V; capillary voltage, 3 kV and cone voltage, 30 V. [Glu1]-Fibrinopeptide B was used as the lock mass generating a reference ion in positive mode. Data were processed employing ProteinLynx Global SERVER 2.5.1 Software (Waters™).

### 4.10. Bioinformatics Analysis of Giardia Genome

*Giardia* genome (https://giardiadb.org/giardiadb/) was surveyed for ABC/MDR transporters homologous sequences. This was performed by using the protein sequences of human P glycoprotein (P-gp/MDR1, GenBank ID AAA59575.1), MRP4 (ABCC4, GenBank ID NP_005836.2) and ABCG2 (GenBank ID AAG52982.1) [14,15,16] as queries. Multiple Blastp searches were performed and alignments were made in BLAST-P (https://blast.ncbi.nlm.nih.gov/Blast.cgi). The sequences found were used for the specific design of primers.

### 4.11. Relative-Quantitative RT-PCR

The expression level of gi*HSP70* and drug transporters (gi*ABC*, gi*ABCP*, gi*MDR*, gi*MRPL* and gi*MDRAP1*) mRNA was determined by relative quantification in real-time PCR (qRT-PCR). Total RNA was obtained from DMSO- or ASA- (0.125, 0.25, 0.5 or 1 mM) treated trophozoites using a Total RNA Purification Kit (Norgen Biotek, Thorold, ON, Canada). cDNA was synthesized by a reverse transcriptase reaction (Verso cDNA Synthesis Kit, Thermo Scientific™) using 1 μg of RNA and Oligo dT primer. This kit is supplied with RT Enhancer to remove contaminating DNA, eliminating the need for DNAse I treatment. Specific primers for each gene were designed and are listed in Table 4. The amplification was performed in a StepOne^TM^ Real-Time PCR System (Applied Biosystems™, Foster, CA, USA) using Maxima SYBR Green/ROX qPCR Master Mix (Thermo Scientific). The expression levels of the above-mentioned genes were normalized to the expression level of shippo1. A melting curve was performed to confirm the absence of contaminants and no dimer formation by the primers. The conditions for RT-qPCR were as follows: hot start at 95 °C for 10 min and 40 cycles of 95 °C for 15 s, 60 °C for 30 s and 72 °C for 30 s. The data were analyzed using StepOne v2.3 Software based on the 2^−ΔΔCt^ method (2^−ΔΔCt^ = RE, relative expression).

### 4.12. ASA Selectivity Index Calculation

#### 4.12.1. Culture of Human Intestinal Caco-2 Cells

The cells were cultured at 37 °C in Dulbecco’s modified Eagle’s culture medium (DMEM), supplemented by 10% fetal bovine serum FBS (Invitrogen), in a humidified atmosphere (5% CO_2_ and 95% air). For routine maintenance, cells were split twice a week by detachment with 0.25% Trypsin-0.025% EDTA (Sigma-Aldrich, Saint Louis, MO, USA), and re-seeding in 25 cm^2^ flasks in a split ratio of 1:4. For experiments, the number of Caco-2 cells per well was estimated by counting the cells in a Neubauer hematocytometer chamber.

#### 4.12.2. Cell Viability (MTT Assay)

Caco-2 cell viability was evaluated by MTT (3-(4,5-dimethylthiazol-2-yl)-2,5-diphenyltetrazolium bromide) tetrazolium reduction assay. The MTT assay was performed with slight modifications as previously described [45]. Briefly, the cells were seeded in cell culture plates of 96 wells at a density of 5000 cells per well and incubated at 37 °C for 48 h. Then cells were treated with DMSO 0.09% and ASA (0.125, 0.25, 0.5 or 1 mM) for 24 h. After the incubation period, the medium was removed and 100 µL of MTT reagent (0.8 mg/mL in serum-free medium) was added to each well, and the cells were incubated at 37 °C for 4 h in 5% CO_2_-95% air atmosphere. Next, the medium was replaced with DMSO (150 μL) to dissolve the formazan crystals and the absorbance was measured at 570 nm with a microplate reader (Bio-Tek Synergy HT).

The selectivity index (SI) was calculated as CC_50_ Caco-2/IC_50_
*Giardia*.

### 4.13. Statistical Analysis

All experiments were performed in triplicate. Data were analyzed by ANOVA and post hoc Dunnett’s multiple comparisons test. *p* values of ≤0.05 were considered statistically significant compared with untreated cells. Error bars in graphics indicate standard deviations for the experiments. IC_50_ calculation was analyzed by nonlinear regression. GraphPad Prism version 6.01 for Windows, GraphPad Software, La Jolla, CA USA was employed.

## Figures and Tables

**Figure 1 pharmaceuticals-13-00440-f001:**
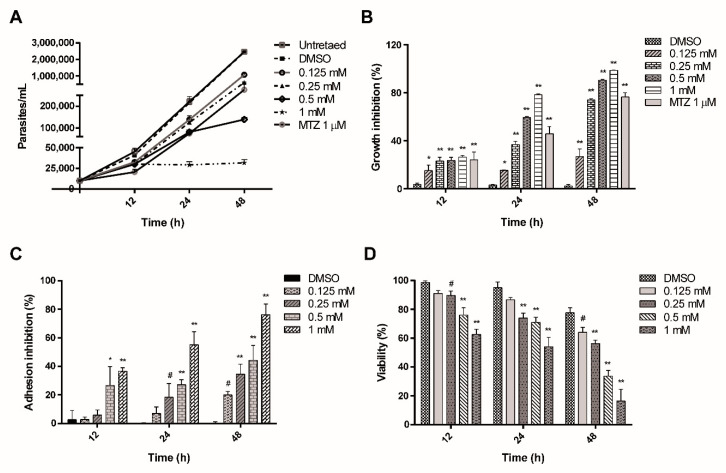
In vitro effect of ASA on *Giardia lamblia* trophozoites after incubation for 12, 24 and 48 h. Growth (**A**), percentage of growth inhibition (**B**), adhesion (**C**) and viability (**D**). The results represent the mean of triplicate determinations ± SD. (* *p* < 0.0006, ^#^
*p* < 0.005, ** *p* < 0.0001).

**Figure 2 pharmaceuticals-13-00440-f002:**
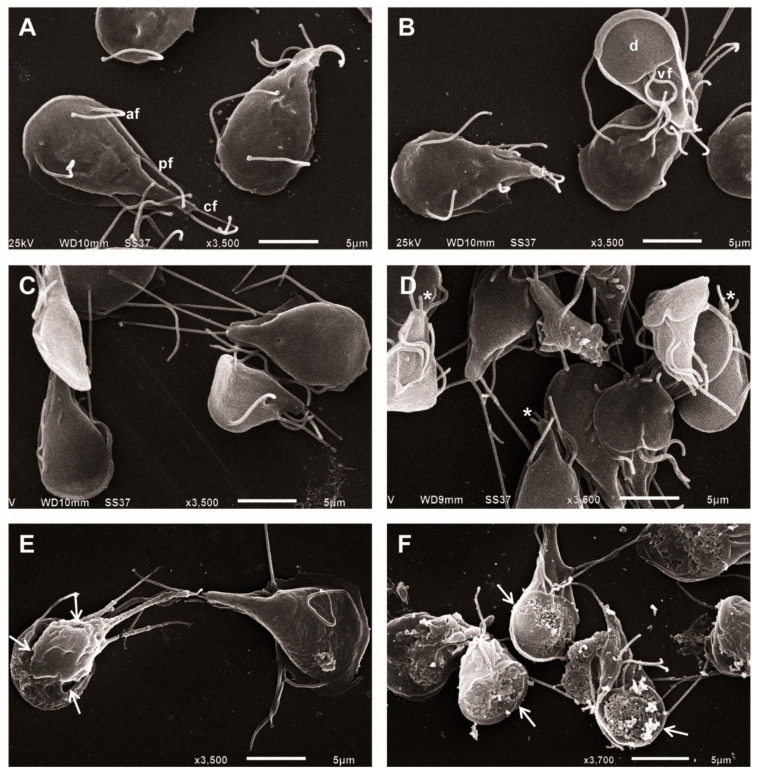
SEM micrographs of *Giardia lamblia* trophozoites after 48 h of treatment with ASA. Untreated cells (**A**), DMSO (**B**), 0.125 mM (**C**), 0.25 mM (**D**), 0.5 mM (**E**) and 1 mM (**F**). (d = ventral disk; af = anterior flagella, vf = ventral flagella, pf = posterior-lateral flagella and cf = caudal flagella). Bar = 5 and 10 μm. Asterisks point flagella shortening and arrows indicate perforations.

**Figure 3 pharmaceuticals-13-00440-f003:**
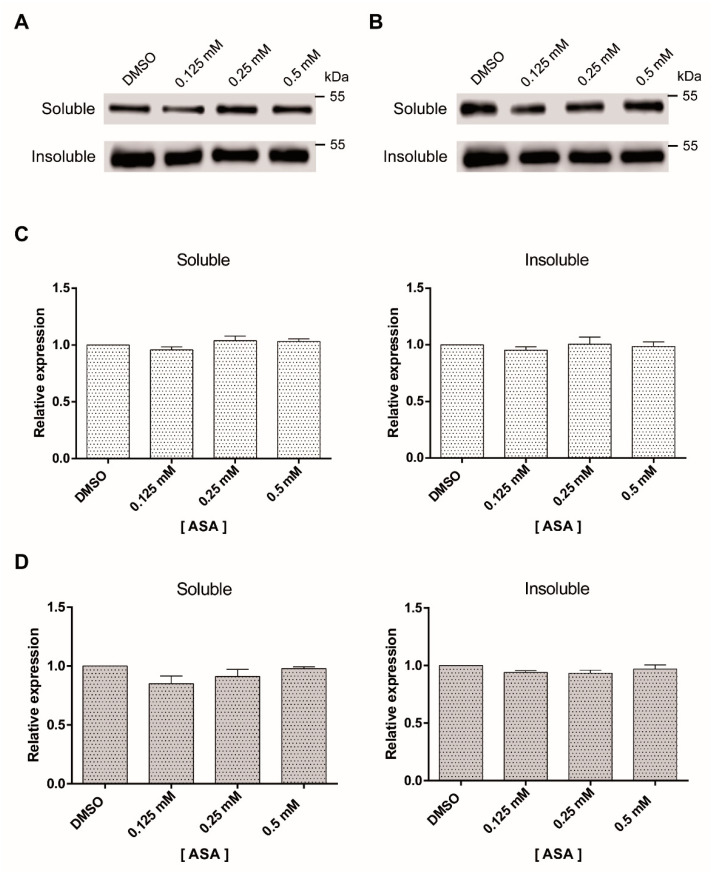
Soluble and insoluble tubulin fractions of *Giardia* trophozoites exposed to ASA. The amount of soluble and insoluble α-tubulin was analyzed by Western blotting after 24 h (**A**) and 48 h (**B**). This experiment was repeated three times. Densitometric analysis of soluble and insoluble tubulin levels after 24 h (**C**) and 48 h (**D**).

**Figure 4 pharmaceuticals-13-00440-f004:**
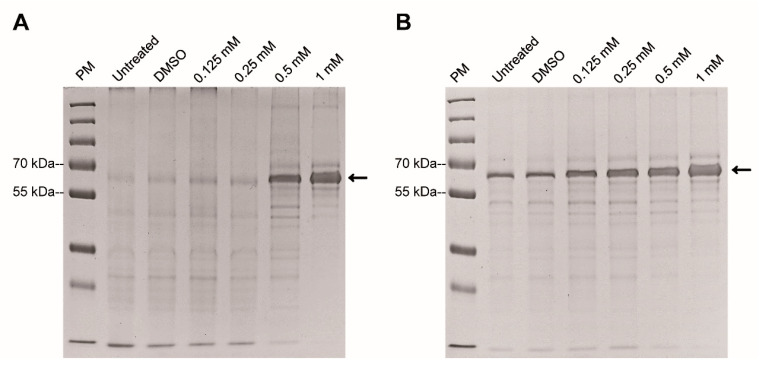
Effect of ASA on electrophoretic protein pattern of *Giardia lamblia*. Trophozoites exposed for 24 h (**A**) and 48 h (**B**). PM, ladder protein marker.

**Figure 5 pharmaceuticals-13-00440-f005:**
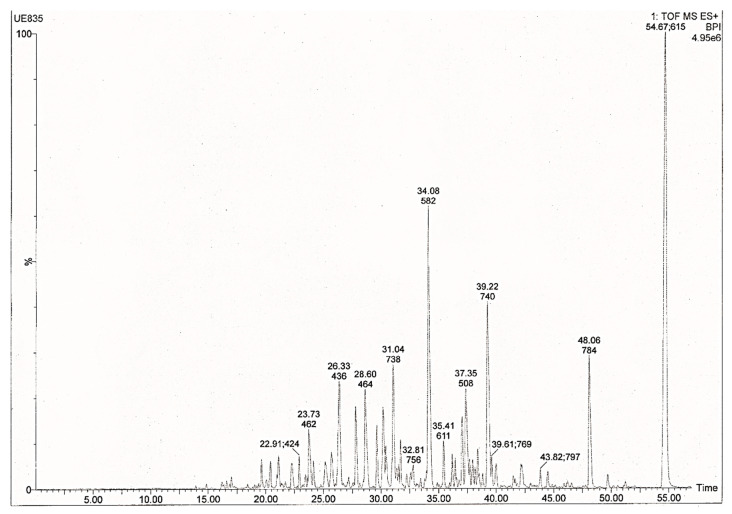
UPLC-TOF mass spectra obtained in a positive ion of tryptic peptides extracted from 55–70 kDa band separated by SDS-PAGE of ASA-treated parasites.

**Figure 6 pharmaceuticals-13-00440-f006:**
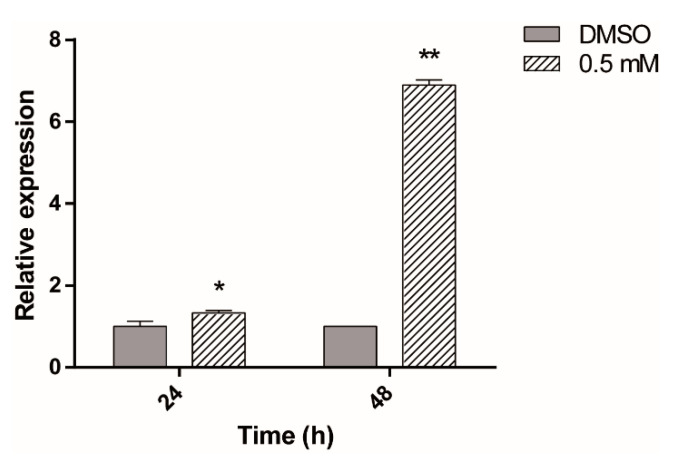
Relative-quantitative RT-PCR assay for *hsp70*. Total RNA was obtained from parasites exposed to DMSO or 0.5 mM of ASA for 24 h and 48 h (* *p* < 0.02, ** *p* < 0.0001).

**Figure 7 pharmaceuticals-13-00440-f007:**
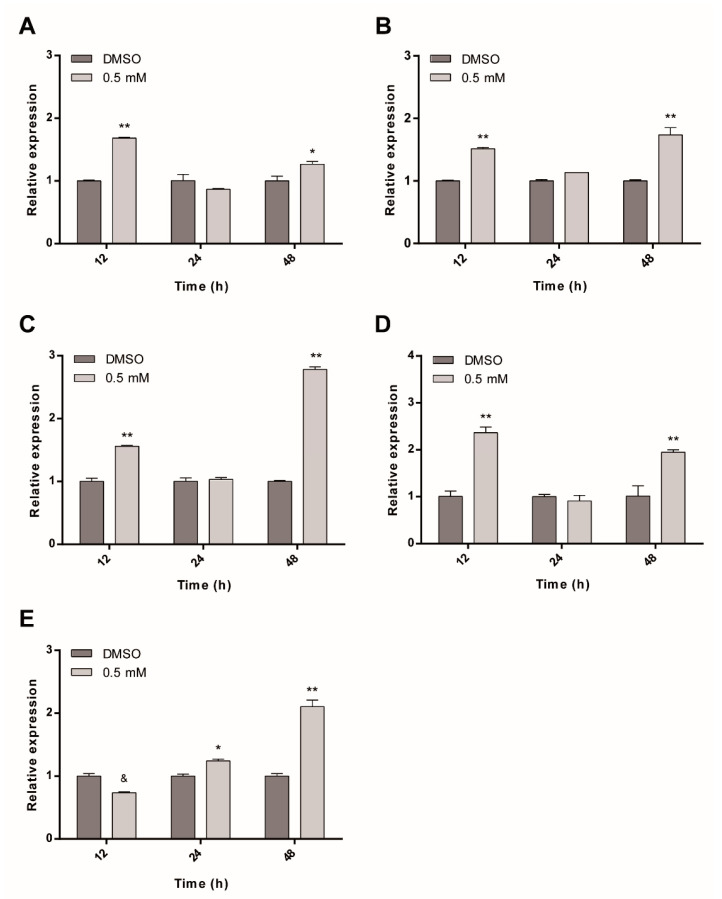
Regulation of expression levels of MDRs genes under ASA treatment. (**A**) giABC. (**B**) giABCP. (**C**) giMDRP. (**D**) giMRPL and (**E**) giMDRAP1. The values are expressed as mean ± SD (*n* = 3). Asterisks indicate statistically significant differences between DMSO control and treatment groups, * *p* < 0.006, ** *p* < 0.0001). In E, giMDR AP1 transcript amount was significantly lower than DMSO control (indicated with &, *p* < 0.004).

**Figure 8 pharmaceuticals-13-00440-f008:**
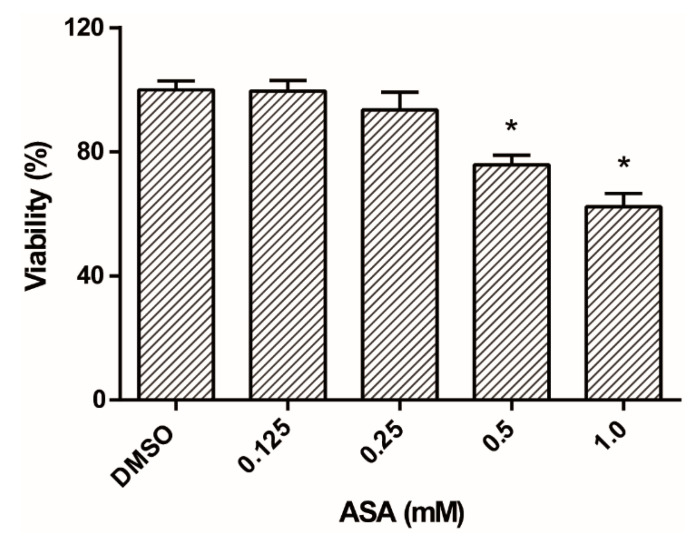
Reduction of Caco-2 cell viability under ASA treatment. * *p* < 0.05.

**Table 1 pharmaceuticals-13-00440-t001:** Antigiardial, cytotoxicity and selectivity index values of acetylsalicylic acid (ASA).

IC_50_ on *Giardia*	CC_50_ on Caco-2 Cells	Selectivity Index CC_50 Caco-2_/IC_50_ *_Giardia_*
0.29 mM	2.32 mM	8

**Table 2 pharmaceuticals-13-00440-t002:** Proteins identified in *Giardia* extract by UPLC-Q-TOF-MS after ASA treatment.

UniProt Access Number	Description	mW (Da)	PLGS Score	Coverage (%)
A8BCR6_GIAIC	Cytosolic HSP70 (GL50803_88765)	71,588	5209.85	49.4
E2RU36_GIAIC	Arginine deiminase (GL50803_112103)	64,090	1510.245	31.4
A8BQS7_GIAIC	HSP90 alpha (GL50803_13864)	36,966	881.7655	24.1
E2RU18_GIAIC	Phosphomannomutase 2 (GL50803_17254)	73,882	792.8965	35.9
E2RTY6_GIAIC	Glucose 6 phosphate isomerase (GL50803_9115)	64,451	691.5236	15.1
A8B2H9_GIAIC	Uncharacterized protein (GL50803_10315)	25,108	627.4312	20.3
A8BBL2_GIAIC	Uncharacterized protein (GL50803_27947)	7457	626.1924	41.5
A8B431_GIAIC	Bip (GL50803_17121)	74,314	586.4369	30.1
A8BUY7_GIAIC	AAA family ATPase (GL50803_16867)	96,298	565.1209	27.5
A8B820_GIAC	Protein 21.1 (GL50803_17060)	66,194	554.8513	27.6
A8BA49_GIAIC	Uncharacterized protein (GL50803_16507)	98,575	517.2396	11.9

**Table 3 pharmaceuticals-13-00440-t003:** Protein BLAST results of multidrug resistance (MDRs) in *Giardia* Data Base.

*Giardia* MDRs	Homo Sapiens MDRs								
	Pgp/MDR1			MRP4			ABCG2		
	Bit Score	Identity (%)	E Value	Bit Score	Identity (%)	E Value	Bit Score	Identity (%)	E Value
giABC							81.6	25	9 × 10^−16^
giABCP	171	24	1 × 10^−42^	360	31	8 × 10^−104^			
giMDRP	140	35	3 × 10^−^^33^						
giMRPL	129	32	2 × 10^−29^	450	34	6 × 10^−133^			
giMDRAP1				533	35	8 × 10^−^^163^			

**Table 4 pharmaceuticals-13-00440-t004:** Specific primers to assess the effect of ASA on the expression level of HSP70 and MDRs by qRT-PCR.

GenBank Accession Number	Gene	Primer (5′-3′)
XM_001707918.1	*HSP70*	*F* 5′-CAT CGC CAA TGA GCA GGG CGC GTA-3′ *R* 5′-ATC GCC CTG TTG CTA CCG GAG A-3′
XM_001706171.1	*Multidrug resistance-associated protein 1* *(giMDRAP1)*	*F* 5′-AGA CTC GAG CGA CAA GAA CCC CAA CCA CGT T-3′ *R* 5′-AGA CTC GAG TGA AGA GCT TGA GGT CGG GTA TC-3′
XM_001705432.1	*ABC transporter family protein* *(giABC)*	*F* 5′-AGA CTC GAG ACA CGA ATA GGT GGT TAG CCG ACT-3′ *R* 5′-AGA CTC GAG AGA CTG ACC CAC ATA TGC CCG C-3′
XM_001709457.1	*MRP-like ABC transporter* *(giMRPL)*	*F* 5′-AGA CTC GAG CAA ATA CAA GTC CAG AGA AGC AGG-3′ *R* 5′-AGA CTC GAG CAG AGA ACC AGT GTC TGT CAA C-3′
XM_001710125.1	*Multidrug resistance ABC transporter ATP-binding and permease protein* *(giMDRP)*	*F* 5′-GAG CTC GAG GGT CTA CTT GAG AAG GCC ATT CCA-3′ *R* 5′-AGA CTC GAG GTC AAC GCT TTT GAA CTT GTG CA-3′
XM_001705989.1	*ABC transporter, putative* *(giABCP)*	*F* 5′-AGA CTC GAG GGG CTT GCC ACA CTT GTT GGC AGC-3′ *R* 5′-AGA CTC GAG GAT GCA CTT GAT AGT CAG AGT CGT-3′
XM_001708537.1	*H-shippo 1*	*F* 5′-CGT CAT CAA CAG GTC CGA-3′ *R* 5′-CCA GCT CTC CTT GAA CAC-3′

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
