# Peer review of "Antigiardial Activity of Acetylsalicylic Acid Is Associated with Overexpression of HSP70 and Membrane Transporters"

_pharmaceuticals, 2020, doi:10.3390/ph13120440_

Round 1

Reviewer 1 Report

The manuscript has been significantly improved and now warrants publication in Pharmaceuticals.

Author Response

We appreciate all your comments and suggestions.

Sincerely,

Dr. Araceli Castillo Romero

Reviewer 2 Report

I appreciate the answers of the authors to my technical points raised in the initial review. However, my major point really is that, based on the data presented, the authors are not able to claim a specific effect of ASA against Giardia. The effective doses that are needed are far beyond general considerations of effective compounds. See for instance the following publication, where screening of libraries was used with a cut-off of 10µM to qualify a potentially effective drug: Tejman-Yarden N, Miyamoto Y, Leitsch D, Santini J, Debnath A, Gut J, et al. A reprofiled drug, auranofin, is effective against metronidazole-resistant Giardia lamblia. Antimicrobial agents and chemotherapy. 2013;57(5):2029-35.

Therefore, I am not able to recommend the manuscript for publication.

Author Response

ANSWER

We value the comments made by Reviewer 2. Our goal for this study was to demonstrate that ASA has antigiardial activity. We show that ASA selectively kills Giardia, and that the drug affects the growth and morphology of trophozoites. It is true that other drugs are effective at lower concentrations like metronidazole (the standard therapy), or auranofin (a potential new antigiardial compound) but, it is important to consider that they are different classes of drugs. Furthermore metronidazole is a prodrug that needs to be activated by anaerobic enzyme systems within the cytosol of Giardia, while auranofin is a gold(I)-containing antirheumatic drug whose activity is associated with its ability to induce heme oxygenase-1 (HO-1). Besides, It is also important to consider that ASA is common Aspirin that is one of the best documented medicines in the world and is one of the most used drugs of all time. It is the most prescribed anti-inflammatory and analgesic-antipyretic agent, and various studies have demonstrated that long-term use of this drug may decrease the risk of cardiovascular disease or various cancers (http://cancer.gov). Depending on the source, the doses range from 75 to 650 mg for pain relieve or cardiovascular disease (Yuxiang Dai and Junbo Ge 2012). As antirheumatic drug Aspirin could be used at higher doses in a range of 0.6 to 4.5 g/day (http://www.arthritis.org), indicating that it is very often well-tolerated drug (Samantha Forder., et al., 2016). Intentional ASA intoxication has been documented with a total drug intake of 60 g (Soleimani R., et al., 2019). As with most drugs Aspirin can causes collateral effects that can be associated with factors including age, doses, and time of drug exposition. In our work, ASA at concentrations of 0.5 and 1 mM (well below concentrations consumed by people) showed better antigiardial activity compared to the most common drug treatment with metronidazole. Thus, ASA shows credible promise as an alternative antigiardial treatment. Our group will expand our studies to investigate the molecular mechanism and agree with Reviewer 2 that other structural modifications of ASA may have improved activity and selectivity. Our manuscript now includes this information in the discussion. See lines 206-208 and 213-219.  

Reviewer 3 Report

Although the current manuscript is full of experiments, they are not coherent and mostly inconclusive. For example, doses of the compound (acetylsalicylic acid or ASA ) is too high, at least ~500 fold higher than metronidazole. Naturally, the question comes if this drug in such a high dose could be effective physiologically. As the authors have demonstrated by qRT-PCR that most of the MDR genes are activated by acetylsalicylate. This is possible because of the high concentration of this compound is triggering their expressions and it doesn't mean that they are the target of acetylsalicylate. Did authors look for COX-1 and COX-2 homologues in Giardia? It is important to run a negative control (in high doses) and a positive control to demonstrate that MDRs are the actual target of ASA. Authors also need to investigate what are the targets of ASA in Caco2 cells.  Are these cells are losing their viability in the presence of the overwhelming concentrations of ASA?  Data showed that the soluble and insoluble tubulins remain unchanged by ASA. What are insoluble tubulins? Are they polymerized microtubules or tubulin associated with membrane component? What are the rationale of using tubulin? Did authors consider using other derivatives of ASA to see the effects on Giardia? 

Author Response

We thank Reviewer 3 for their critical comments, and we have edited our manuscript to reflect their concerns.

First, ASA has been in use since 1899, it is the most prescribed anti-inflammatory and analgesic-antipyretic agent (see our response above to Reviewer 1 comments). Several studies have demonstrated the antimicrobial activity of this drug (Wang WH., et al., 2003; A G Al-Bakri., et al., 2009).  Our group is the first report that ASA may be a good candidate to consider as possible antigiardial drug. We demonstrated a significant decrease in viability and cellular damage in membrane, ventral disk, flagella, and caudal region within 24 h in comparison to metronidazole. Like Reviewer 2, Reviewer 3 also had concerns for the dosing needed to kill Giardia (please see below de response to reviewer 2).

[Response to reviewer 2: We value the comments made by Reviewer 2. Our goal for this study was to demonstrate that ASA has antigiardial activity. We show that ASA selectively kills Giardia, and that the drug affects the growth and morphology of trophozoites. It is true that other drugs are effective at lower concentrations like metronidazole (the standard therapy), or auranofin (a potential new antigiardial compound) but, it is important to consider that they are different classes of drugs. Furthermore metronidazole is a prodrug that needs to be activated by anaerobic enzyme systems within the cytosol of Giardia, while auranofin is a gold(I)-containing antirheumatic drug whose activity is associated with its ability to induce heme oxygenase-1 (HO-1). Besides, It is also important to consider that ASA is common Aspirin that is one of the best documented medicines in the world and is one of the most used drugs of all time. It is the most prescribed anti-inflammatory and analgesic-antipyretic agent, and various studies have demonstrated that long-term use of this drug may decrease the risk of cardiovascular disease or various cancers (http://cancer.gov). Depending on the source, the doses range from 75 to 650 mg for pain relieve or cardiovascular disease (Yuxiang Dai and Junbo Ge 2012). As antirheumatic drug Aspirin could be used at higher doses in a range of 0.6 to 4.5 g/day (http://www.arthritis.org), indicating that it is very often well-tolerated drug (Samantha Forder., et al., 2016). Intentional ASA intoxication has been documented with a total drug intake of 60 g (Soleimani R., et al., 2019). As with most drugs Aspirin can causes collateral effects that can be associated with factors including age, doses, and time of drug exposition. In our work, ASA at concentrations of 0.5 and 1 mM (well below concentrations consumed by people) showed better antigiardial activity compared to the most common drug treatment with metronidazole. Thus, ASA shows credible promise as an alternative antigiardial treatment. Our group will expand our studies to investigate the molecular mechanism and agree with Reviewer 2 that other structural modifications of ASA may have improved activity and selectivity. Our manuscript now includes this information in the discussion. See lines 206-208 and 213-219].

On the other hand, as we mentioned in the manuscript, previous reports have shown that ASA has activity on human Pgp/MDR1, MRP4, and ABCG2 genes, this effect has been shown to occur in Caco-2 cells (line 239). Our evidence demonstrated that, as in mammal cells, ASA induces changes of ABC transporters MDR1 expression in Giardia trophozoites, may be acting as a self-defense system. We agree with Reviewer 3 that experiments are necessary to confirm ABC transporters as targets of ASA, however, our goal was to demonstrate the antigiardial activity of ASA.

We have edited our text to explain the difference between soluble (cytosolic) and insoluble tubulin (polymerized) (line 290). The analysis of these fractions was supported for the fact that ASA provoked morphological alterations in trophozoites, and tubulin is one of the major components of cytoskeleton (Kari D Hagen., et al., 2020).

We hope that we have clarified the concerns of the reviewers and that our paper will be suitable for publication.  Thank you for your consideration.

Sincerely,

Dr. Araceli Castillo Romero

Round 2

Reviewer 2 Report

I thank the authors for their kind reply. I see the points raised by the authors, however, I admit that I am just not convinced that the authors show a specific effect of ASA on Giardia parasites in the current manuscript. In my view the authors likely describe a non-specific off-target effect by ASA. They also provide no evidence for a particular molecular target of ASA in Giardia. The latter is of course a difficult task, however, in spite of the high doses of ASA necessary for the effects described it is in my view essential to exclude off-target effects. It is certainly disappointing, but in light that the first reviewer was more positive I suggest that the editors ask a third party reviewer for an additional scientific oppinion.

Author Response

Guadalajara, Jalisco, México November 24, 2020.

Response to Reviewer 2 Comment

Comment:

I thank the authors for their kind reply. I see the points raised by the authors; however, I admit that I am just not convinced that the authors show a specific effect of ASA on Giardia parasites in the current manuscript. In my view the authors likely describe a non-specific off-target effect by ASA. They also provide no evidence for a particular molecular target of ASA in Giardia. The latter is of course a difficult task, however, in spite of the high doses of ASA necessary for the effects described it is in my view essential to exclude off-target effects. It is certainly disappointing, but in light that the first reviewer was more positive I suggest that the editors ask a third party reviewer for an additional scientific opinion.

ANSWER

We value the comments made by Reviewer 2. But we consider that our evidence shows a specific effect doses-time dependent of ASA on Giardia growth and morphology. The non-cytotoxic effects observed by ASA in Caco-2 cells support the selectivity of the drug against this parasite. With respect to the molecular target, we agree that it will improve the impact of our work, however, our goal for this study was to demonstrated that ASA could be a promising new antigiardial drug or a starting point for structural modifications in the search for new antigiardial treatments. About the action mechanism, our group is currently addressing that item. It is also important to mention that at this moment there are no published reports about ASA in Giardia. Moreover, this is the first report showing that ASA kills Giardia, and as in mammal cells, the effect of the drug was accompanied by variations in the ABC transporters and HSP70 mRNA expression levels.

Sincerely

Araceli Castillo Romero

Reviewer 3 Report

No specific comments.

Author Response

We appreciate the reviewer’s thoughtful suggestions, which help to improve the quality of our manuscript. Even though there are no specific comments, we updated the conclusions as requested by the reviewer, please see conclusion section lanes 245-246.  We hope you will now find our manuscript suitable for publication.